# RFMEDSAM 2: AUTOMATIC PROMPT REFINEMENT FOR MEDICAL IMAGE SEGMENTATION WITH SAM 2

## ABSTRACT

Segment Anything Model 2 (SAM 2) is a prompt-driven foundation model that extends SAM to both image and video domains, demonstrating superior zero-shot performance over its predecessor. While SAM 2 builds on SAM's success in medical image segmentation, it retains limitations such as binary mask outputs, lack of semantic label inference, and reliance on precise prompts for target object identification. Moreover, applying SAM and SAM 2 directly to medical image segmentation tasks often yields suboptimal results. In this paper, we investigate the upper performance limit of SAM 2 using custom fine-tuning adapters and ground-truth prompts, achieving a Dice Similarity Coefficient (DSC) of 92.30% on the BTCV dataset Landman et al. (2015), surpassing the state-of-the-art nnUNet by 12%. To address prompt dependency, we explore multiple prompt generation strategies and introduce a UNet that autonomously predicts masks and bounding boxes, which are then used as input to SAM 2. Dual-stage refinements within SAM 2 further improve performance. Extensive experiments demonstrate that our method achieves state-of-the-art results on the AMOS2022 Ji et al. (2022) dataset, with a 1.4% Dice improvement over nnUNet, and outperforms nnUNet by 6.4% on the BTCV dataset Landman et al. (2015).

## 1 INTRODUCTION

Medical image segmentation is essential for biomedical analysis, aiding in disease diagnosis, anomaly detection, and surgical planning. Deep learning-based recently approaches Ronneberger et al. (2015); Isensee et al. (2019); Zhou et al. (2021) have significantly advanced segmentation tasks, with convolutional neural networks (CNNs) and vision transformers (ViTs) emerging as dominant architectures. However, medical imaging datasets often suffer from a scarcity of high-quality annotations,

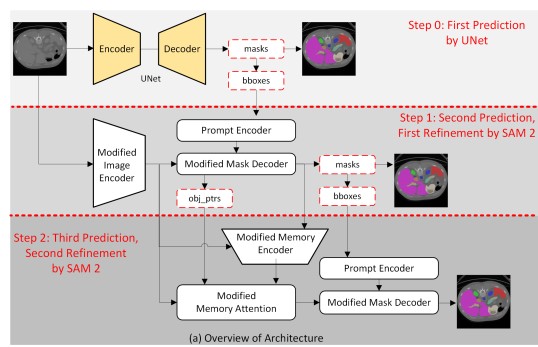

Figure 1: Overview of our proposed RFMedSAM 2.

which hampers the training of large-scale models. Consequently, architectures with strong inductive biases, such as CNNs, have been more feasible to train from scratch for medical segmentation tasks.

Foundation models Devlin et al. (2018); He et al. (2022), trained on vast datasets, have demonstrated remarkable zero-shot and few-shot generalization across diverse applications OpenAI (2023); Radford et al. (2021). These models have shifted the paradigm from training task-specific models from scratch to a "pre-training then fine-tuning" approach, significantly impacting computer vision. The introduction of the Segment Anything Model (SAM) Kirillov et al. (2023), trained on the SA-1B dataset, marked a breakthrough in prompt-driven natural image segmentation. SAM's success has extended to various applications, including medical image segmentation Ma et al. (2024); Xie et al. (2024; 2025); Deng et al. (2023); Zhang & Liu (2023); Bui et al. (2024).

Building on this, SAM 2 was introduced as an enhanced version of SAM, extending its functionality to both image and video domains. SAM 2 enables real-time segmentation across video sequences using a single prompt. Tab. 1 shows that SAM 2 outperforms SAM on the BTCV dataset Landman

et al. (2015), achieving a Dice score of 82.77% compared to SAM's 81.89%, highlighting its potential for further exploration in medical image segmentation tasks.

However, like SAM, SAM 2 has inherent limitations, including its binary mask outputs, the absence of semantic label inference, and dependence on precise prompts for target object identification. Additionally, without modifications, the performance of SAM and SAM 2 on medical segmentation tasks remains suboptimal compared to state-of-the-art models.

To address these challenges and maximize SAM 2's potential for medical segmentation, we contribute:

- We introduce RFMedSAM 2, a novel framework for automatic prompt refinement in medical image segmentation, leveraging the multi-stage refinement capabilities of SAM 2.
- We develop new adapter modules: a depth-wise convolutional adapter (DWConvAdapter) for attention blocks and a CNN-Adapter for convolutional layers, enhancing spatial information capture and enabling efficient fine-tuning.
- We establish the upper performance bound of SAM 2 with optimal prompts, achieving a DSC of 92.30%, surpassing the state-of-the-art nnUNet by 12% on BTCV Landman et al. (2015) dataset.
- We propose an independent UNet to generate masks and bounding boxes for SAM 2, enabling automatic prompt generation and dual-stage refinement, eliminating reliance on manual prompts.
- We conduct extensive experiments on challenging medical image datasets (AMOS Ji et al. (2022) and BTCV Landman et al. (2015)), demonstrating that RFMedSAM 2 achieves state-of-the-art results, surpassing nnUNet by 1.4% on the AMOS2022 dataset and 6.4% on the BTCV dataset.

## 2 RELATED WORK

### 2.1 MEDICAL IMAGE SEGMENTATION

The field of medical image segmentation has evolved significantly, with deep learning-based approaches replacing traditional machine learning methods. U-Net Ronneberger et al. (2015) remains a foundational model due to its encoder-decoder structure and skip connections, which help preserve spatial context. Building on this, nnUNet Isensee et al. (2019) introduced an automated pipeline that adapts U-Net's architecture to different medical datasets. Other convolution-based methods, including 3D-UXNET Lee et al. (2022), MedNeXt Roy et al. (2023), and STU-Net Huang et al. (2023), have further advanced segmentation capabilities. More recently, transformer-based models, such as UNETR Hatamizadeh et al. (2022), SwinUNETR Hatamizadeh et al. (2021), and nnFormer Zhou et al. (2021), have been explored to capture global context and improve accuracy by leveraging self-attention mechanisms, which facilitate long-range dependency modeling. While these models have been explicitly designed for medical image segmentation and trained from scratch, they exhibit high inductive bias, which can limit adaptability.

### 2.2 SAM AND SAM 2 FOR MEDICAL IMAGE SEGMENTATION

Segment Anything Model (SAM) Kirillov et al. (2023), pre-trained on over 1 billion masks from 11 million natural images, has emerged as a powerful prompt-based foundation model for image segmentation, demonstrating strong zero-shot capabilities across diverse applications. Following the "pre-training then fine-tuning" paradigm, SAM has been extended and fine-tuned for medical image segmentation in several studies, including MedSAM Ma et al. (2024), MaskSAM Xie et al. (2024), and Self-Prompt SAM Xie et al. (2025), among others Zhang et al. (2024); Deng et al. (2023); Ma & Wang (2023); Wu et al. (2023); Li et al. (2023); Gong et al. (2023). These adaptations highlight SAM's flexibility and the research community's ongoing efforts to tailor it for medical applications. However, SAM's original design limitations, including binary mask outputs and prompt dependency, restrict its effectiveness for fully automated medical segmentation tasks, which are inherited to SAM 2 and make it less suitable for fully automated medical segmentation tasks. SAM2-Adapter Chen et al. (2024) integrates adapters into the image encoder and fine-tunes the mask decoder, yet it still struggles with semantic labels and requires additional prompts, limiting its effectiveness. Similarly, Polyp SAM 2 Mansoori et al. (2024) and other studies Yu et al. (2024); Liu et al. (2024) have explored SAM 2 for medical applications but face the same fundamental challenges.

The dependency on accurate prompts in SAM 2 and other prompt-driven models remains a key limitation, particularly in medical imaging, where obtaining precise annotations can be difficult.

| Prompt | Bounding boxes as prompts | | | | | | Central points as prompts | | | | | | |
|---|---|---|---|---|---|---|---|---|---|---|---|---|---|
| Method | SAM | SAM 2 | | | | | | SAM | SAM 2 | | | | | |
| # frames / class | All | All | | Two | | One | | All | All | | Two | | One | |
| Frames for Step 2 | – | All | Unprompted | All | Unprompted | All | Unprompted | – | All | Unprompted | All | Unprompted | All | Unprompted |
| DSC (%) | 81.89 | 81.17 | 82.77 | 68.75 | 68.03 | 45.00 | 44.07 | 8.86 | 3.81 | 4.90 | 2.11 | 3.43 | 2.53 | 4.59 |

Table 1: Performance evaluation of SAM and SAM 2 with different prompt settings on BTCV dataset.

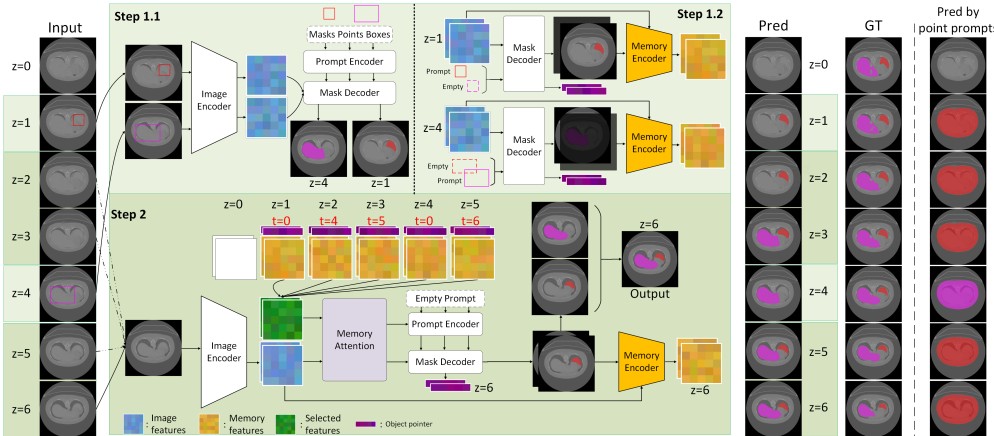

Figure 2: Overview of SAM 2. The pipeline includes steps for prompted and unprompted frames.

Ongoing research seeks to mitigate this issue through auxiliary models that generate reliable prompts and learning mechanisms that dynamically refine prompts during training. While SAM 2's memory attention improves temporal consistency in video segmentation, it also introduces additional complexity in training and memory requirements. Addressing these challenges could enable SAM 2 and similar models to reach their full potential in medical image segmentation, bridging the gap between state-of-the-art performance and practical usability.

## 3 THE PROPOSED APPROACH

This section details the structure and functionality of our proposed method. We begin by an overview and analysis of SAM 2, and introduce RFMedSAM 2, describing its architectural innovations and strategies for prompt refinement, which enables RFMedSAM 2 to achieve state-of-the-art performance in medical image segmentation while reducing its reliance on precise manual prompts. Fig. 2 illustrates the architecture and pipeline of SAM 2.

### 3.1 OVERVIEW OF SAM AND SAM 2

**Model Architecture.** Both SAM and SAM 2 share a core structure consisting of an image encoder, a prompt encoder, and a mask decoder. The image encoder processes input images to generate embeddings, while the prompt encoder handles input prompts in the form of points, bounding boxes, or masks. The mask decoder then combines image and prompt embeddings to produce binary segmentation masks. SAM employs a Vision Transformer as the backbone of its image encoder, whereas SAM 2 utilizes Hiera Ryali et al. (2023) to enhance feature representation. Additionally, SAM 2 introduces a memory attention module that conditions current frame features on past frames and object pointers, along with a memory encoder that fuses current frame features with output masks to generate memory features.

**SAM 2's Pipeline.** The pipeline of SAM 2 operates in two stages: i) Prompted Frame Processing (Step 1.1 and Step 1.2 in Fig. 2), where segmentation is guided by explicit prompts. In this stage, SAM 2 segments objects in frames that contain explicit user-defined or ground-truth prompts. Each frame is processed independently, using the given prompt to generate an object mask. To ensure comprehensive segmentation, the number of processed instances is dynamically adjusted based on the number of expected objects. The resulting predicted masks and object pointers are then stored in the memory encoder to generate memory features for subsequent frames. ii) Unprompted Frame Processing, where memory attention propagates cues from prior frames as implicit prompts (Step 2 in Fig. 2). This stage handles frames without explicit prompts by leveraging temporal information from previously segmented frames. The memory attention module aggregates features from past prompted and unprompted frames to provide contextual cues for segmenting the current frame. Prompted frames are assigned a temporal position of 0, while unprompted frames receive increasing temporal

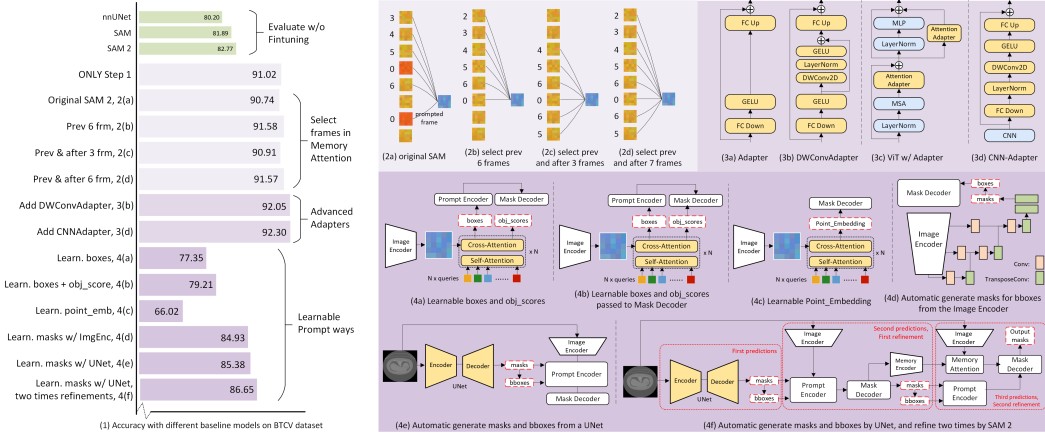

Figure 4: (1) Performance comparisons based on proposed methods. (2) Ablation studies on frame selection strategies. (3) Proposed Adapters. (4) Ablation studies on prompt generators.

positions (up to 6), with more recent frames being weighted more heavily. This approach enables continuity in segmentation but may introduce errors if temporal positioning is not accurately aligned.

## 3.2 ANALYSIS AND INSIGHTS

Tab. 1 summarizes experiments on BTCV with various settings for SAM and SAM 2 using ground-truth prompts and no structural changes. For each frame where an object (class) appears, we only use one prompt. As the official SAM 2 allows up to two prompted frames per object during training, we report results with one and two prompts per class, and also include the "all

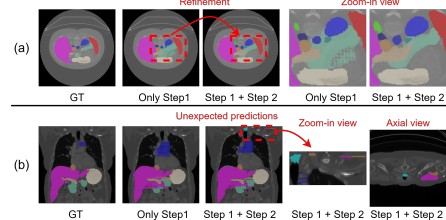

Figure 3: Benefits of refinement in Step 2.

prompted frames" setting for fair comparison with SAM, which requires prompts for every frame. For step 2 in SAM 2, we test two strategies: i) the official protocol (step 2 only for unprompted frames), and ii) forcely applying it to all frames (including prompted ones) to assess refinement potential.

- **i) Bounding box prompts vs. central points:** As shown in Tab. 1, using central point prompts results in a Dice score below 10% for both SAM and SAM 2, indicating their ineffectiveness for segmentation. In contrast, bounding box prompts significantly improve performance. Due to this, all subsequent experiments utilize bounding boxes as prompts.
- **ii) Per-frame prompts:** The results indicate that SAM 2 achieves its highest performance (82.77% Dice) when each frame contains a bounding box for every object, highlighting the critical role of per-frame prompts in ensuring optimal accuracy.
- **iii) Comparison between SAM and SAM 2:** With per-frame prompts, SAM achieves a Dice score of 81.89%, whereas SAM 2 improves upon this, reaching 82.77%. This demonstrates the performance enhancement offered by SAM 2 over its predecessor.
- **iv) Step 2 for refinement:** Step 2 in SAM 2, which leverages memory attention for unprompted frames, can also be applied to all frames for refinement. Enforcing Step 2 across all frames results in a slight drop in Dice score from 82.77% to 81.17%, yet it demonstrates potential for refining segmentation results, as illustrated in Fig. 3. Fig. 3(a) highlights an example where the green area is refined for better accuracy. However, Fig. 3(b) shows a limitation: assigning all prompted frames a temporal position of 0 can lead to incorrect temporal positioning, causing false positives in unrelated frames. Correcting temporal position can prevent such errors.
- **v) Streaming operation:** Most SAM 2 modules operate on 2D image frames independently to reduce memory cost, while the memory attention module aggregates features from previous and prompted frames to establish temporal context. This streaming design balances efficiency and effectiveness, so we retain it in our method (Fig. 4).

## 3.3 RFMEDSAM 2 ARCHITECTURE

RFMedSAM 2 is a refined adaptation of SAM 2 designed to enhance segmentation performance for medical imaging tasks. Fig. 1 illustrates the overall architecture, which consists of three sequential stages: an initial prediction stage, a preliminary segmentation stage, and a refinement stage.

In the initial prediction stage, a U-Net processes medical images to generate multi-class masks. These masks are converted into bounding boxes that serve as prompts for subsequent stages. The preliminary segmentation stage integrates the modified SAM 2 framework, where the image encoder extracts embeddings from input images, and the prompt encoder transforms bounding boxes into point embeddings. The mask decoder then utilizes these embeddings to generate initial masks and object pointers, which are refined into updated bounding box prompts. The modified memory encoder further processes these masks and frame features to generate memory features.

The final refinement stage enhances segmentation accuracy by leveraging a modified memory attention module, which establishes spatial and temporal relationships by combining the current frame's image features with memory features from previous frames. The mask decoder processes these integrated features along with new point embeddings from the prompt encoder, producing refined segmentation predictions as the final output.

### 3.3.1 ARCHITECTURAL ADAPTATIONS TO SAM 2

To optimize SAM 2 for medical image segmentation, RFMedSAM 2 incorporates targeted modifications to address modality differences, spatial complexity, and the need for temporal consistency.

The image encoder is adapted to accommodate multi-channel medical images while maintaining compatibility with SAM's RGB-based input expectations. Two stacked convolutional layers are introduced to transform medical images into the expected input format while preserving spatial details. The Hiera Ryali et al. (2023) backbone is enhanced with Depth-wise Convolutional Adapters (DWConvAdapters) in attention blocks and CNN-Adapters in the FPN module. DWConvAdapters apply depth-wise convolutions separately to each input channel, reducing computational load while improving spatial feature extraction, which is crucial for anatomical segmentation. CNN-Adapters further refine feature fusion in convolutional layers, enhancing segmentation accuracy.

The mask decoder is modified to improve spatial learning. Adapters are positioned after self-attention and cross-attention blocks and in parallel with MLP layers to improve feature representation. DWConvAdapters strengthen the decoder's ability to capture fine-grained spatial details, while CNN-Adapters optimize feature adaptation across different anatomical structures.

Further modifications enhance temporal consistency by refining the U-Net, memory encoder, and memory attention mechanisms. The U-Net maintains a symmetric encoder-decoder structure with skip connections to retain spatial information. CNN-Adapters in the memory encoder refine feature processing, ensuring that previously segmented frames contribute effectively to future predictions. DWConvAdapters are integrated into the memory attention module to enhance spatial-temporal feature aggregation.

### 3.3.2 FRAME SELECTION AND MEMORY ATTENTION STRATEGY

SAM 2's memory attention mechanism plays a crucial role in propagating segmentation information across frames. However, its original temporal positioning strategy assigns a temporal position of zero to all prompted frames, leading to ambiguity and reduced segmentation accuracy. To address this, we explored alternative temporal positioning strategies, as summarized in Fig. 4(2).

The baseline SAM 2 strategy (Fig. 4(2a)) achieved a Dice Similarity Coefficient (DSC) of 90.74%, but was outperformed by a more structured approach. Our improved strategy (Fig. 4(2b)) assigns a temporal position of zero to only the current frame, incorporating up to six preceding frames with progressively higher temporal positions. This refined strategy enhances segmentation consistency by distinguishing current from previous frames, achieving a DSC of 91.58%. Alternative strategies incorporating forward and backward frame selection (Figures 4(2c)-(2d)) either reduced performance or increased memory overhead, confirming the effectiveness of our approach.

Additionally, RFMedSAM 2 maintains the streaming operation proposed in SAM 2, where most modules process images independently, reducing memory usage. The memory attention module remains the only component that integrates contextual information from previous frames, making it both efficient and effective for sequential medical image segmentation.

### 3.3.3 NOVEL ADAPTERS FOR ENHANCED FINE-TUNING

To support parameter-efficient fine-tuning while preserving SAM 2's zero-shot capabilities, RFMed-SAM 2 introduces novel adaptation mechanisms that enhance spatial and convolutional processing.

DWConvAdapters (Depth-Wise Convolutional Adapters) are incorporated into the image encoder, memory attention, and mask decoder to improve spatial feature extraction. By applying depth-wise convolutions separately to each channel, these adapters reduce computational complexity while maintaining fine-grained spatial detail, a critical factor in segmenting anatomical structures. Experimental results demonstrate that integrating DWConvAdapters led to a 0.47% improvement in DSC, validating their effectiveness in spatial learning (Fig. 4(3b)).

CNN-Adapters further refine feature adaptation in convolutional layers, particularly in the FPN module and memory encoder. These adapters optimize multi-scale feature representation, ensuring robust segmentation performance across diverse medical imaging datasets. Their inclusion resulted in a 0.25% increase in DSC, confirming their impact on segmentation accuracy (Fig. 4(3b)).

The final RFMedSAM 2 model integrates DWConvAdapters for image embedding attention blocks, CNN-Adapters for convolutional layers, and original adapters for point embedding attention blocks. These enhancements collectively yield a 4% improvement in segmentation performance over state-of-the-art methods, as shown in Tab. 2. This demonstrates that our fine-tuning strategy effectively enhances SAM 2's adaptability for complex medical imaging tasks.

## 3.4 ENHANCING PROMPT GENERATION

Accurate ground truth (GT) prompts enable SAM 2 to achieve state-of-the-art segmentation performance, yet their reliance on precise, manually annotated prompts limits practical deployment in real-world medical imaging. Manually generating high-quality prompts for every frame is labor-intensive and inconsistent, making an automated, self-sufficient prompt generation mechanism essential. To bridge this gap, we designed a prompt generation framework that progressively refines both the generated prompts and the final segmentation outputs during training.

Our objective is to replace explicit GT prompts with automatically generated ones while maintaining high segmentation accuracy. As illustrated in Fig. 4(4a)-(4f), we explored six distinct prompt generation strategies, categorized into two main types: learnable point coordinate representations (Figures 4(4a)-(4c)) and learnable masks (Figures 4(4d)-(4f)). Their effectiveness is summarized in the last six bars of Fig. 4(1). Below, we analyze both approaches and explain the rationale for our final design choice.

### 3.4.1 LEARNABLE POINT COORDINATE REPRESENTATIONS

One intuitive approach is to learn point coordinates directly. The block depicted in Fig. 4(4a) initializes object queries for each class, processing them through self-attention and cross-attention mechanisms that interact with current image features. Multiple MLP layers adjust embedding dimensions to generate box coordinates and object scores. Unlike its predecessor, SAM 2 applies stricter labeling criteria for point prompts, classifying them as no object (-1), negative/positive points (0,1), or box prompts (2,3). Previous experiments using GT prompts included labels indicating the absence of objects in certain frames. In our approach, object scores are trained to determine whether a frame should contain a prompt.

Despite these efforts, directly learning point coordinates proved challenging. The model in Fig. 4(4a) achieved only a DSC of 77.35%, revealing a substantial performance gap. To improve accuracy, we incorporated object scores from the mask decoder (Fig. 4(4b)), leading to a 1.9% improvement. However, performance remained suboptimal. An alternative strategy involved using a learnable point embedding block (Fig. 4(4c)), where coordinate and label representations were directly learned, resulting in an 11% drop, indicating that learning precise prompt coordinates from scratch is unreliable.

The core issue lies in the difficulty of predicting coordinates without inherent spatial context. Image embeddings lack coordinate encoding, and their initialization is random, making it difficult for the model to align spatially meaningful prompts. Furthermore, bounding boxes alone fail to capture the semantic richness required for robust multi-class segmentation. These limitations led us to shift toward learnable mask-based prompting strategies.

| Semantic labels | Prompts | Method | Spl. | R.Kd | L.Kd | GB | Eso. | Liver | Stom. | Aorta | IVC | Panc. | RAG | LAG | Duo. | Blad. | Pros. | Average |
|---|---|---|---|---|---|---|---|---|---|---|---|---|---|---|---|---|---|---|
| ✔ | – | UNETR Hatamizadeh et al. (2022) | 0.928 | 0.913 | 0.903 | 0.719 | 0.763 | 0.955 | 0.849 | 0.922 | 0.838 | 0.766 | 0.663 | 0.663 | 0.662 | 0.815 | 0.744 | 0.807 |
| | | nnFormer Zhou et al. (2021) | 0.950 | 0.948 | 0.944 | 0.789 | 0.784 | 0.967 | 0.914 | 0.931 | 0.868 | 0.828 | 0.654 | 0.695 | 0.759 | 0.865 | 0.773 | 0.845 |
| | | SwinUNETR Hatamizadeh et al. (2021) | 0.954 | 0.954 | 0.950 | 0.819 | 0.852 | 0.972 | 0.919 | 0.955 | 0.911 | 0.875 | 0.775 | 0.801 | 0.816 | 0.895 | 0.812 | 0.884 |
| | | SwinUNETRv2 ? | 0.959 | 0.962 | 0.958 | 0.842 | 0.867 | 0.976 | 0.933 | 0.957 | 0.920 | 0.889 | 0.783 | 0.812 | 0.843 | 0.913 | 0.836 | 0.897 |
| | | 3D UX-Net Lee et al. (2022) | 0.955 | 0.956 | 0.953 | 0.826 | 0.858 | 0.972 | 0.922 | 0.955 | 0.915 | 0.881 | 0.781 | 0.809 | 0.820 | 0.902 | 0.823 | 0.889 |
| | | nn-UNet Isensee et al. (2019) | 0.951 | 0.961 | 0.956 | 0.826 | 0.869 | 0.973 | 0.931 | 0.957 | 0.923 | 0.880 | **0.784** | 0.809 | 0.846 | 0.898 | 0.827 | 0.893 |
| ✗ | nnUNet | SAM Kirillov et al. (2023) bbox | 0.679 | 0.741 | 0.640 | 0.168 | 0.443 | 0.773 | 0.671 | 0.651 | 0.554 | 0.434 | 0.232 | 0.324 | 0.444 | 0.698 | 0.602 | 0.538 |
| ✗ | nnUNet | SAM 2 Ravi et al. (2024) bbox | 0.784 | 0.817 | 0.819 | 0.664 | 0.734 | 0.780 | 0.697 | 0.793 | 0.739 | 0.536 | 0.457 | 0.604 | 0.563 | 0.744 | 0.691 | 0.695 |
| ✗ | nnUNet | MedSAM Ma et al. (2024) bbox | 0.714 | 0.811 | 0.702 | 0.193 | 0.469 | 0.759 | 0.725 | 0.701 | 0.681 | 0.434 | 0.365 | 0.412 | 0.462 | 0.783 | 0.758 | 0.600 |
| ✔ | No needs | SAMed Zhang & Liu (2023) | 0.849 | 0.857 | 0.830 | 0.573 | 0.733 | 0.894 | 0.816 | 0.855 | 0.784 | 0.727 | 0.622 | 0.683 | 0.701 | 0.844 | 0.819 | 0.772 |
| ✔ | No needs | SAM3D Bui et al. (2024) | 0.796 | 0.863 | 0.871 | 0.428 | 0.711 | 0.908 | 0.833 | 0.878 | 0.749 | 0.699 | 0.564 | 0.607 | 0.635 | 0.884 | 0.840 | 0.751 |
| ✔ | No needs | **RFMedSAM 2** | **0.972** | **0.971** | **0.966** | **0.887** | **0.878** | **0.980** | **0.943** | **0.958** | **0.925** | **0.896** | 0.781 | **0.811** | **0.853** | **0.921** | **0.859** | **0.907** |

Table 2: Comparison of RFMedSAM 2 with SOTA methods on AMOS testing dataset by Dice Score.

### 3.4.2 LEARNABLE MASKS

Rather than learning point coordinates, we explored a more structured approach: generating segmentation masks first and deriving bounding boxes from them. The architecture shown in Fig. 4(4d) follows a hierarchical design that integrates convolutional layers with multi-scale features from the image encoder. It starts from lower-resolution feature maps and progressively refines them through convolutional layers, incorporating higher-resolution details at each stage. The generated masks are supervised by auxiliary loss functions that compare them against ground truth labels, achieving a DSC of 84.93% - a noticeable improvement over the learnable point coordinate approaches.

However, this method introduced training challenges. The auxiliary losses from generated masks sometimes conflicted with the final segmentation losses from SAM 2, making it difficult to optimize both components simultaneously. Additionally, the architectural disparity between the prompt generator and SAM 2 led to synchronization issues, hindering the convergence during training.

To mitigate these conflicts, we introduced an independent U-Net architecture for mask generation alongside SAM 2 (Fig. 4(4e)). This ensured that SAM 2's parameter updates remained unaffected by the prompt generation process. The U-Net-generated masks were converted into bounding boxes and used as input prompts for SAM 2, increasing DSC to 85.38%.

To further refine interactions between U-Net and SAM 2, we incorporated a multi-stage refinement process: generated masks and bounding boxes were first passed to Step 1 of SAM 2, producing an initial set of refined segmentations. These refined outputs were then fed into Step 2, enabling further enhancement. This achieved a final DSC of 86.48%, validating the pipeline's effectiveness.

## 4 EXPERIMENTAL EVALUATION

### 4.1 DATASETS AND EVALUATION METRICS

We conducted experiments using two publicly available datasets: the AMOS22 Abdominal CT Organ Segmentation dataset Ji et al. (2022) and the Beyond the Cranial Vault (BTCV) challenge dataset Landman et al. (2015). **(i)** The AMOS22 dataset consists of 300 abdominal CT scans with manual annotations for 16 anatomical structures, serving as the basis for multi-organ segmentation tasks. The test set includes 200 images, and our model is evaluated using the AMOS22 leaderboard. **(ii)** The BTCV dataset comprises 30 cases of abdominal CT scans. Following established split strategies Hatamizadeh et al. (2021), we use 24 cases for training and 4 cases for validation. Performance is assessed using the average Dice Similarity Coefficient (DSC) across 13 abdominal organs.

In Tables 2 and 3, "Semantic labels" indicate a model's ability to infer and predict labels, while "Prompt" specifies the source of prompts. Since SAM and MedSAM do not predict semantic labels and require additional prompts, we use GT or predictions inferred by a pre-trained nnUNet to generate prompts, with the corresponding labels used as semantic labels.

### 4.2 COMPARISON WITH STATE-OF-THE-ART METHODS

### 4.2.1 RESULTS ON THE AMOS22 DATASET

Tab. 2 presents the quantitative results on the AMOS22 dataset, comparing our proposed RFMedSAM 2 with widely recognized segmentation methods, including CNN-based approaches (nnUNet Isensee et al. (2019)), transformer-based models (UNETR Hatamizadeh et al. (2022), SwinUNETR Hatamizadeh et al. (2021), nnFormer Zhou et al. (2021)), and SAM-based methods (SAM Kirillov et al. (2023), SAM 2 Ravi et al. (2024), MedSAM Ma et al. (2024), SAMed Zhang &

| Semantic labels | Prompts | Method | Spl. | R.Kd | L.Kd | GB | Eso. | Liv. | Stom. | Aorta | IVC | Veins | Panc. | AG | DSC |
|---|---|---|---|---|---|---|---|---|---|---|---|---|---|---|---|
| ✔ | – | TransUNet Chen et al. (2021) | 0.952 | 0.927 | 0.929 | 0.662 | 0.757 | 0.969 | 0.889 | 0.920 | 0.833 | 0.791 | 0.775 | 0.637 | 0.838 |
|  |  | 3D UX-Net Lee et al. (2022) | 0.946 | 0.942 | 0.943 | 0.593 | 0.722 | 0.964 | 0.734 | 0.872 | 0.849 | 0.722 | 0.809 | 0.671 | 0.814 |
|  |  | UNETR Hatamizadeh et al. (2022) | 0.968 | 0.924 | 0.941 | 0.750 | 0.766 | 0.971 | 0.913 | 0.890 | 0.847 | 0.788 | 0.767 | 0.741 | 0.856 |
|  |  | Swin-UNETR Hatamizadeh et al. (2021) | 0.971 | 0.936 | 0.943 | 0.794 | 0.773 | 0.975 | 0.921 | 0.892 | 0.853 | 0.812 | 0.794 | 0.765 | 0.869 |
|  |  | nnUNet Isensee et al. (2019) | 0.942 | 0.894 | 0.910 | 0.704 | 0.723 | 0.948 | 0.824 | 0.877 | 0.782 | 0.720 | 0.680 | 0.616 | 0.802 |
|  |  | nnFormer Zhou et al. (2021) | 0.935 | 0.949 | 0.950 | 0.641 | 0.795 | 0.968 | 0.901 | 0.897 | 0.859 | 0.778 | 0.856 | 0.739 | 0.856 |
| ✗ | GT | SAM Kirillov et al. (2023) | 0.933 | 0.922 | 0.927 | 0.805 | 0.831 | 0.899 | 0.808 | 0.890 | 0.894 | 0.492 | 0.728 | 0.708 | 0.819 |
| ✗ | GT | SAM 2 Ravi et al. (2024) | 0.946 | 0.923 | 0.924 | 0.859 | 0.888 | 0.928 | 0.893 | 0.852 | 0.884 | 0.434 | 0.694 | 0.705 | 0.828 |
| ✗ | GT | MedSAM Ma et al. (2024) | 0.751 | 0.814 | 0.885 | 0.766 | 0.721 | 0.901 | 0.855 | 0.872 | 0.746 | 0.771 | 0.760 | 0.705 | 0.803 |
| ✗ | GT | SAM-U Deng et al. (2023) | 0.868 | 0.776 | 0.834 | 0.690 | 0.710 | 0.922 | 0.805 | 0.863 | 0.844 | 0.782 | 0.611 | 0.780 | 0.790 |
| ✗ | GT | SAM-Med2D Cheng et al. (2023) | 0.873 | 0.884 | 0.932 | 0.795 | 0.790 | 0.943 | 0.889 | 0.872 | 0.796 | 0.813 | 0.779 | 0.797 | 0.847 |
| ✗ | GT | RFMedSAM 2 | 0.961 | 0.943 | 0.945 | 0.909 | 0.918 | 0.965 | 0.945 | 0.954 | 0.942 | 0.968 | 0.883 | 0.843 | 0.923 |
| ✔ | No Needs | SAMed Zhang & Liu (2023) | 0.862 | 0.710 | 0.798 | 0.677 | 0.735 | 0.944 | 0.766 | 0.874 | 0.798 | 0.775 | 0.579 | 0.790 | 0.776 |
| ✔ | No Needs | SAM3D Bui et al. (2024) | 0.933 | 0.901 | 0.909 | 0.601 | 0.733 | 0.944 | 0.882 | 0.856 | 0.778 | 0.722 | 0.759 | 0.590 | 0.801 |
| ✔ | No Needs | RFMedSAM 2 | 0.969 | 0.947 | 0.953 | 0.611 | 0.817 | 0.974 | 0.909 | 0.917 | 0.887 | 0.803 | 0.865 | 0.747 | 0.867 |

Table 3: Comparison of RFMedSAM 2 with state-of-the-art methods on the BTCV dataset. "Semantic labels" indicate the model's ability to infer labels, while "Prompts" specify the source of the prompt.

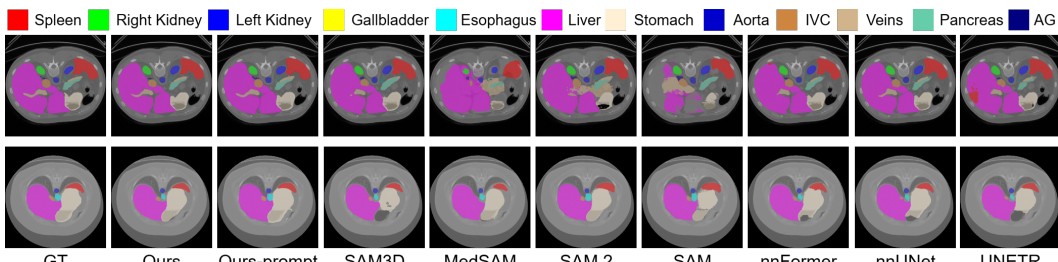

Figure 5: Qualitative comparison on the BTCV dataset. RFMedSAM 2 provides the most precise segmentation for each class and exhibits fewer segmentation outliers.

Liu (2023), and SAM3D Bui et al. (2024)). To ensure a fair comparison, all methods are evaluated using 5-fold cross-validation without ensemble.

We observe that RFMedSAM 2 outperforms all existing methods on most organs, achieving a new state-of-the-art performance in DSC. When using nnUNet-generated bounding box prompts, SAM, SAM 2, and MedSAM exhibit DSC decreases of 34%, 18%, and 27%, respectively, compared to nnUNet's accuracy of 89.3%. These reductions highlight the limitations of relying on external prompt sources. Among SAM-based models, SAM 2 achieves the best performance, demonstrating its strong zero-shot capabilities.

Notably, RFMedSAM 2 surpasses nnUNet by 1.4% in DSC and outperforms SAMed and SAM3D by 23% and 25%, respectively. This significant improvement confirms that our proposed prompt-free RFMedSAM 2 outperforms other prompt-free SAM models. On the highly challenging AMOS22 dataset, RFMedSAM 2 achieves state-of-the-art performance, validating our method's effectiveness.

### 4.2.2 RESULTS ON THE BTCV DATASET

Tab. 3 presents the quantitative performance on the BTCV dataset, comparing RFMedSAM 2 with leading SAM-based methods using proper prompts (*i.e.*, SAM Kirillov et al. (2023), SAM 2 Ravi et al. (2024), MedSAM Ma et al. (2024), SAM-U Deng et al. (2023), and SAM-Med2D Cheng et al. (2023)), SAM-based methods without prompts (*i.e.*, SAMed Zhang & Liu (2023) and SAM3D Bui et al. (2023)), convolution-based methods (VNet Ronneberger et al. (2015) and nnUNet Isensee et al. (2019)), and transformer-based methods (TransUNet Chen et al. (2021), SwinUNet Cao et al. (2021), and nnFormer Zhou et al. (2021)).

RFMedSAM 2 outperforms all existing methods, establishing a new state-of-the-art benchmark. When provided with proper prompts, RFMedSAM 2 achieves a Dice Similarity Coefficient (DSC) of 92.3%, marking a substantial 5% improvement over the previous best-performing method. Among SAM-based methods using proper prompts, SAM-Med2D achieves the highest DSC of 84.7%, which RFMedSAM 2 surpasses by 7.6%, highlighting its superior effectiveness in leveraging prompts.

In prompt-free settings, RFMedSAM 2 outperforms the other prompt-free SAM-based methods, surpassing SAMed and SAM3D by 9% and 6%, respectively. Compared to non-SAM-based methods, RFMedSAM 2 exceeds nnUNet and nnFormer by 6.4% and 1% in DSC, demonstrating its capability even on highly saturated datasets. Fig. 5 provides qualitative comparisons, illustrating that RFMedSAM 2 predicts the labels for 'Stomach,' 'Spleen,' and 'Liver' with greater accuracy.

|  | train with prompts | learnable bboxes | learnable masks |
|---|---|---|---|
| w/ obj_score | 0.923 | 0.792 | 0.847 |
| w/o obj_score | 0.920 | 0.628 | 0.867 |

Table 4: Performance of different models with and without object score prediction on BTCV dataset.

| Dataset | Step 0 - UNet | Step 1 - SAM | Step 2 - SAM |
|---|---|---|---|
| BTCV | 0.856 | 0.864 | 0.867 |
| AMOS | 0.895 | 0.898 | 0.907 |

Table 5: Performance of output predictions across different steps. Two refinements.

|  | (2, 1024, 1024) | (8, 512, 512) | (32, 256, 256) |
|---|---|---|---|
| DSC | 0.751 | 0.827 | 0.867 |

Table 6: Performance comparison of different patch sizes on the BTCV dataset.

|  | 3D UNet | 2D UNet | 2D UNet + Attention | 3D UNet + Attention |
|---|---|---|---|---|
| DSC | 0.825 | 0.807 | 0.805 | 0.815 |

Table 7: Performance comparison of different UNet models on the BTCV dataset.

### 4.3 ANALYSIS

**Refinements.** Tab. 5 presents experimental results for output predictions at different steps on the BTCV and AMOS datasets. The results demonstrate a gradual improvement in performance, starting from the initial prediction at Step 0 (UNet), followed by refinement at Step 1 (SAM 2), and further refinement at Step 2 (SAM 2). Fig. 6 visualizes these comparisons across the three steps, illustrating how segmentation gaps are progressively filled through the two refinement stages, underscoring the effectiveness of our model's refinement process.

A standalone UNet with the same structure as Step 0 achieves only 82.5% DSC, a 4.2% drop in performance. Training jointly with SAM 2 improves UNet's performance due to loss propagation, where SAM 2 effectively acts as a teacher model, refining UNet's feature representations.

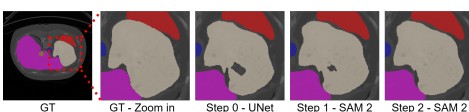

Figure 6: Comparison of Step 0, 1, and 2.

We evaluate three baseline models: fine-tuning SAM 2 with prompts, using learnable bounding boxes as the prompt generator, and using learnable masks as the prompt generator, both with and without object score prediction. Tab. 4 provides the results, revealing the following insights:

- Learning object scores with prompts does not significantly enhance performance compared to using prompts without object scores, as the presence of a prompt inherently implies the object's existence in a given frame.
- The model with learnable bounding boxes benefits from object score learning since bounding box predictions often exhibit lower accuracy.
- The model with learnable masks performs worse when incorporating object scores. This is likely because the predicted masks already capture a more informative probability distribution, whereas object scores impose a single probability estimate, potentially reducing accuracy.

**Input Patch Sizes and UNet Architectures.** Tab. 6 evaluates the effect of different input patch sizes while maintaining a constant total number of pixels. Increasing the number of depth slices improves performance, highlighting the benefits of capturing volumetric information. Tab. 7 compares various UNet architectures, showing that 3D UNet outperforms 2D UNet due to its ability to learn depth-wise features. However, incorporating attention blocks in the bottleneck does not yield improvements, likely due to the strong inductive biases present in medical image segmentation tasks.

## 5 CONCLUSION

In this paper, we present RFMedSAM 2, a framework for automatic prompt refinement that extends SAM 2 with multiple refinement stages for volumetric medical image segmentation. First, we evaluated SAM 2's upper performance bound with accurate prompts. To enhance spatial feature extraction and enable efficient fine-tuning, we introduced depth-wise convolutional adapters for attention blocks and CNN-Adapters for convolutional layers, along with optimized memory attention positioning. These improvements yielded a DSC of 92.3%, surpassing nnUNet by 12% on BTCV Landman et al. (2015). Second, we eliminated reliance on manual prompts by designing an independent U-Net to generate masks and bounding boxes as inputs to SAM 2, followed by two refinement stages. This achieved DSCs of 90.7% on AMOS2022 Ji et al. (2022) and 86.7% on BTCV. Overall, RFMedSAM 2 achieves state-of-the-art segmentation performance, and future work will explore extensions to MRI, ultrasound, and real-time clinical applications.

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

# A   INTRINSIC ISSUES OF SAM2

Figure 2 illustrates the whole pipeline of SAM 2, highlighting several intrinsic issues for medical image segmentation.

i) **Omission to predict the first few frames:** The first frame in two objects is the second frame, therefore, SAM 2 begins processing from the second frame, disregarding the first frame, even though it contains both objects.

ii) **Empty prompt affecting object prediction:** When no prompt is provided for an object, but the object is still present, the empty prompt restricts prediction for that object. For instance, at frames $z = 1$ and $z = 4$, the purple and red objects, respectively, are omitted from the predictions.

iii) **Confusion of temporal positions:** All prompted frames are assigned a temporal position of 0. While this approach increases attention to the prompted frames, it loses the relative temporal positioning of all the prompt frames. Moreover, since SAM 2 skips over prompted frames, the relative temporal positions of the unprompted frames are distorted. For example, the real relative temporal position of the frame $z = 3$ with respect to the current frame $z = 6$ should be 3, but due to the prompted frame at $z = 4$, the relative temporal position is incorrectly assigned as 2.

# B   POTENTIALS TO FORCE STEP 2 FOR ALL FRAMES.

When we provide prompts at each frame for each class, SAM 2 does not process Step 2 and does not leverage the capabilities of Memory Attention, which can build relations with previous frames and prompted frames. To explore this functionality, we force Step 2 for all frames after processing Step 1. Although the results decreased slightly from 82.77% to 81.17% Dice, we find a potential

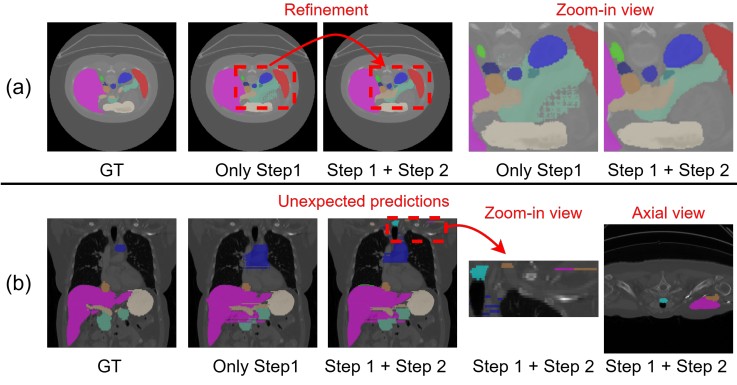

Figure 7: Benefits for refinement by Step 2.

refinement benefit illustrated in Figure 7(a). Through Step 2, the green area is refined and becomes more accurate, demonstrating the significant potential of the refinement process. As a result, we plan to incorporate this approach into our method. However, the refinement introduced by Step 2 also has some drawbacks. In Figure 7(b), we show orthogonal planes in relation to the axial plane (a sequence of the axial plane images is fed to SAM 2). The top portion of Figure 7(b) presents unexpected predictions. Since SAM 2 assigns a temporal position of 0 to the prompted frames, which are always involved in memory attention, the incorrect relative temporal positioning leads to these unexpected and incorrect predictions. We will address this issue in the next section.

# C   MOTIVATION BEHIND THE DESIGNED ADAPTERS.

Since the image encoder, the memory attention, and the mask decoder contain attention blocks for image embedding, which includes significant spatial information. Therefore, we design the depth-wise convolutional adaption (DWConvAdapter) illustrated in Figure 4(3b) to learn spatial information. After using DWConvAdapters for the attention blocks with image embedding, the performance increases by 0.47%. The motivation behind the DWConvAdapter design is to extend the original adapter by incorporating a depth-wise convolution layer, followed by layer normalization

Figure 8: Details of the whole architecture of RFMedSAM 2.

and a GeLU activation function, to effectively learn spatial information. A parallel skip connection is included to preserve the original structure. In the worst case, where the depth-wise layer learns nothing (*i.e.,* its output is zero), the skip connection ensures that all original information is retained. Building on this concept, we designed the CNN-Adapter for adapting convolutional layers since more convolutional layers are involved at SAM 2 compared to SAM. The CNN-Adapter uses a point-wise convolutional layer to downsample the channel dimension, reducing complexity, followed by a depth-wise convolutional layer to capture spatial dimensions. Finally, a point-wise convolutional layer recovers the channel dimension to its original size. Inspired by ConvNext, we use only layer normalization and a GeLU activation function in this block. The bottleneck structure helps reduce complexity, and a parallel skip connection ensures that the output from the convolutional layers in SAM 2 is preserved. In the worst case, where the depth-wise layer learns nothing (*i.e.,* its output is zero), the skip connection still retains all relevant information.

# D   ARCHITECTURE OF RFMEDSAM 2

Figure 8(a) illustrates the overall pipeline and architecture of RFMedSAM 2, which consists of three primary steps. In Step 0, an additional UNet model is employed to take medical images as input, generating initial multi-class mask predictions, which are then used to create auxiliary bounding boxes for the prompt requirements of SAM 2. In Step 1, the medical images being input are involved into a modified image encoder to produce image embeddings, while the prompt encoder processes the auxiliary bounding boxes to generate point embeddings. These embeddings are passed to the modified mask decoder to generate masks and object pointers. The generated masks are then employed to create second bounding boxes for Step 2. A modified memory encoder processes both the generated masks and current frame features to produce memory features for the next step. Step 2 presents the second prediction by refining the initial predictions and performing the first refinement. In Step 3, the same image features from the modified image encoder are input into a modified memory attention module, which establishes relationships with memory features from previous frames. The output from this memory attention mechanism is fed into the modified mask decoder, while the memory decoder also processes new point embeddings from the prompt encoder. Step 3 generates the third set of predictions and the second refinement, with the final mask prediction being output by the mask decoder. Figure 8(b)-(e) illustrates each component of RFMedSAM 2, described as follows.

## D.1   MODIFIED IMAGE ENCODER

Figure 8(b) illustrates the redesigned image encoder. i) SAM works on natural images that have 3 channels for RGB while medical images have varied modalities as channels. There are gaps between the varied modalities of medical images and the RGB channels of natural images. Therefore, we design a sequence of two stacked convolutional layers to an invert-bottleneck architecture to learn the adaption from the varied modalities with any size to 3 channels. ii) SAM 2 employs Hiera Ryali et al. (2023) that is hierarchical with multiscale output features as its image encoder backbone and a FPN module. Hiera consists of four stages with different feature resolutions and every stage contains various number of attention blocks. We insert our designed DWConvAdapter blocks into each attention block in Hiera. The output of each stage will be connected with one convolution in the FPN module. The latest output feature is up-sampled and summed with the second latest output feature as the image embedding. The third and fourth latest output feature are as skip connections

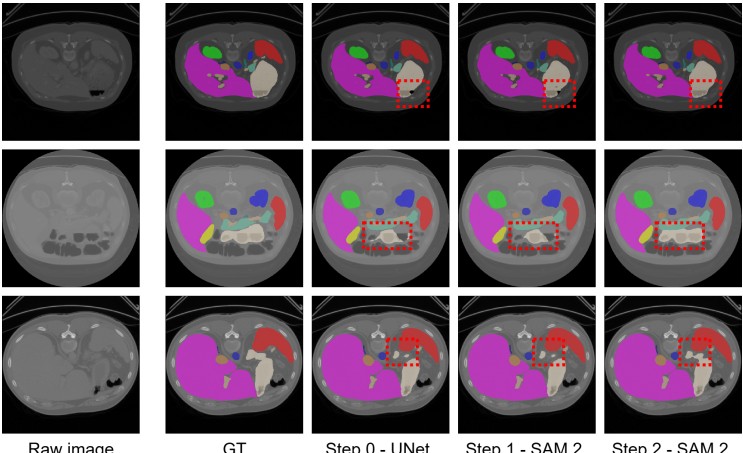

Raw image      GT      Step 0 - UNet      Step 1 - SAM 2      Step 2 - SAM 2

Figure 9: More visualization of two refinements.

to to incorporate high-resolution embeddings for the mask decoding. To adapt these convolutional layers, we insert our designed CNN-Adapters for the output features from the FPN module.

### D.2    MODIFIED MASK ENCODER

Figure 8(c) illustrates the redesigned mask encoder. The mask encoder contains two subsequent transformers and two following convolutional layers. i) Each transformer first applies self-attention to the prompt embedding. We insert an adapter behind the self-attention. Then, a cross-attention block is adopted for tokens attending to image embedding. We insert an adapter behind the cross-attention. Next, we insert a adapter parallel to an MLP block. Finally, a cross-attention block is utilized for image embedding attending to tokens. We insert a DWConvAdapter behind the cross-attention. In this way, our model can learn the spatial information for the image embedding and adapt information for the prompt embedding. ii) We inserted a CNN-Adapter behind the two following convolution layers to adapt convolutional layers from natural images to medical images.

### D.3    UNET, MODIFIED MEMORY ENCODER AND MODIFIED MEMORY ATTENTION.

Figure 8(d) and (e) illustrate the UNet and the redesigned memory encoder, respectively. i) UNet is designed with a symmetrical encoder-decoder structure with skip connections. The encoder consists of several stages, each formed by a sequence of convolutional layers followed by down-sampling layers, progressively increasing the number of channels while reducing the spatial resolution to capture different deep-level features. The decoder upsamples the feature maps using transposed convolutions to restore spatial resolution and refine predictions. Skip connections between corresponding encoder and decoder layers enable the network to retain fine-grained spatial details, enhancing localization accuracy. ii) The memory encoder comprises two modules: the mask downsampler, which processes predicted masks, and the fuser, which integrates image features and mask features. To adapt these CNN-based modules to medical images, a CNN-Adapter is inserted after each module. iii) The memory attention module stacks several transformer blocks, the first one taking the image encoding from the current frame as input. Each block performs self-attention, followed by cross-attention to memory features. Therefore, we inserted our designed DWConvAdapter blocks into each attention block since the transformer blocks process the image embedding with the spatial dimension.

## E    IMPACT OF AUXILIARY LOSSES ON IMAGE ENCODER PARAMETER UPDATES IF PROMPT GENERATOR BUILT WITH IMAGE ENCODER

Figure 4(4d) illustrates a hierarchical structure with convolutional layers combined with multi-level features from the image encoder. The features with a lower resolution gradually increase the resolution by convolution layers and then combined with higher resolution features. Auxiliary loss functions are employed to supervise between the predicted masks and the ground truth. Although this approach achieves a DSC of 84.93%, the result is not competitive. During training, both the auxiliary losses

from the generated masks and the final output losses from SAM 2 influence the update of the image encoder parameters, which constitute a significant portion of the model. However, these two types of losses, due to their distinct architectural differences, are challenging to optimize simultaneously and achieve a balanced update for the image encoder parameters.

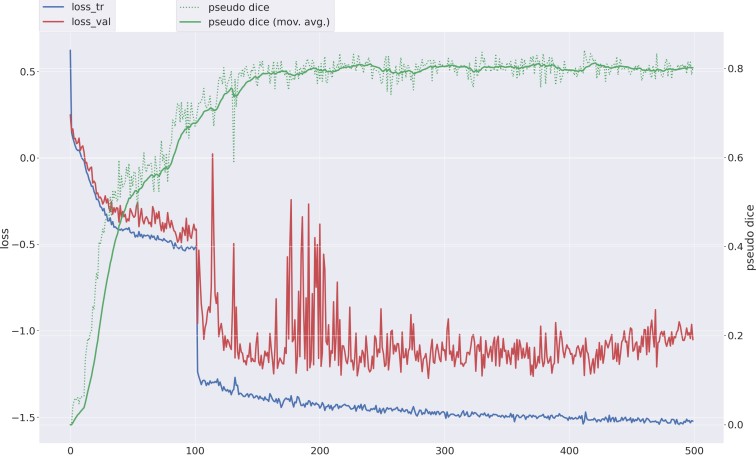

Figure 10: Oscillated losses if prompt generator built with image encoder.

We conduct experiments to validate the insights presented in Figure 10. The training process is divided into two phases: one phase updates the parameters based solely on the auxiliary losses supervised by the auxiliary loss function, while the other phase updates all parameters based on both the auxiliary loss function and the final output loss function. The results indicate that after the second phase begins, the validation loss oscillates and is in an unstable state shown in the red line. The dice of the auxiliary masks present an unstable state since the final output losses affect the update of the image encoder and then affect the accuracy of the auxiliary masks.

In conclusion, using a prompt generator built with the image encoder creates a challenge in balancing the update of the image encoder's parameters. As a result, we abandon this approach and instead employ an independent U-Net to generate masks and subsequently produce the corresponding bounding boxes.

## F  ANALYSIS OF PARAMETERS

| Method | UNet | Adapters | Total Trainable Params | SAM 2 | Total Params |
|--------|------|----------|------------------------|-------|--------------|
| RFMedSAM 2 | 46M | 19M | 46M+19M=65M | 224M | 65M+224.4M=289.4M |

Table 8: The parameters of each components for our RFMedSAM 2.

Table 8 illustrates the number of parameters for each components. Our method is based on the SAM 2 large model, which contains 224.4M parameters in total. These parameters are entirely frozen during training. On top of SAM 2, we introduce two trainable components: i) A UNet (step 0) with 46M parameters. ii) Our designed adapters, inserted into SAM 2, with 19M parameters. Together, these components result in 65M trainable parameters, which is approximately 29% of the total model size (65M out of 224.4M + 65M = 289.4M, shown at the table below). Despite training only a small fraction of the overall parameters, our method achieves efficient adaptation and delivers state-of-the-art performance.

## G  IMPLEMENTATION DETAILS

We utilize some data augmentations such as rotation, scaling, Gaussian noise, Gaussian blur, brightness, and contrast adjustment, simulation of low resolution, gamma augmentation, and mirroring. We set the initial learning rate to 0.001 and employ a "poly" decay strategy in Eq. equation 1.

$$lr(e) = init\_lr \times (1 - \frac{e}{\text{MAX\_EPOCH}})^{0.9}, \qquad (1)$$

where $e$ means the number of epochs, MAX_EPOCH means the maximum of epochs, set it to 1000 and each epoch includes 250 iterations. We utilize SGD as our optimizer and set the momentum to 0.99. The weighted decay is set to 3e-5. We utilize both cross-entropy loss and dice loss by simply summing them up as the loss function. We utilize instance normalization as our normalization layer. we employ the deep supervision loss for the supervision of the U-Net. All experiments are conducted using two NVIDIA RTX A6000 GPUs with 48GB memory.

**Deep Supervision.** The U-Net network is trained with deep supervision. For each deep supervision output, we downsample the ground truth segmentation mask for the loss computation with each deep supervision output. The final training objective is the sum of all resolutions loss:

$$\mathcal{L} = w_1 \cdot \mathcal{L}_1 + w_2 \cdot \mathcal{L}_2 + w_3 \cdot \mathcal{L}_3 + \cdots w_n \cdot \mathcal{L}_n \tag{2}$$

where the weights halve with each decrease in resolution (*i.e.,* $w_2 = \frac{1}{2} \cdot w_1$; $w_3 = \frac{1}{4} \cdot w_1$, etc), and all weight are normalized to sum to 1. Meanwhile, the resolution of $\mathcal{L}_1$ is equal to $2 \cdot \mathcal{L}_2$ and $4 \cdot \mathcal{L}_3$.

## H   MORE VISUALIZATION OF TWO REFINEMENTS

In Figure 9, we present additional qualitative results showcasing the refinements at different stages. With the two refinements, the results clearly illustrate the progressive improvement in segmentation accuracy, emphasizing the effectiveness of our model's refinement process.

