# OpenReview forum: "RFMedSAM 2: Automatic Prompt Refinement for Enhanced Volumetric Medical Image Segmentation with SAM 2"
_ICLR.cc/2026/Conference — ICLR 2026 Conference Withdrawn Submission_

### Official Review · Reviewer_BiLJ · 2025-10-31

**Soundness:** 2
**Presentation:** 1
**Contribution:** 1
**Rating:** 2
**Confidence:** 5

**Summary:**

This paper proposes a medical image segmentation framework that uses SAM2 as a segment refiner. In this paper, SAM2 is employed to further refine the segmentation results of nn-UNet, but the performance improvement is not significant.

**Strengths:**

1.This paper provides detailed explanations in certain sections, particularly regarding the SAM2 framework's workflow and the ablation experiments.
2.The research motivation of this paper is clear.

**Weaknesses:**

1.The structure of the paper is unclear. The methods section mixes in a large portion of the ablation study, such as the Frame Selection Strategies and the Prompt Generation sections. The methods section should clearly and explicitly describe the methods used, while other experimental attempts should be  part of the ablation study in the experiments section.
2.The overall method lacks detailed illustration. There should be a clear diagram illustrating how the proposed method processes 3D images. Figure 2 only shows the workflow for processing a single 2D image, while other components are described separately in different sections. For example, Figure 4 outlines the Frame Selection Strategy and Prompt Generation, but the method framework in Figure 4 (4f) still shows the workflow for processing a single 2D image. The separate illustrations create significant difficulties for readers in understanding the overall methodology of the paper.
3.The training strategy is unclear. The paper lacks necessary training details of components such as UNet and DWConvAdapter.
4.Most importantly, the performance improvement brought by the proposed method is not significant. As shown in Table 5, despite incorporating a considerable number of components and processes, the proposed method only improves performance by about 1% over the  nn-UNet used in Step 0.

**Questions:**

When the amount of data is insufficient to train a robust baseline network (like the nnU-Net used in this paper), does the method proposed in this paper could achieve better refinement results?

---

### Official Review · Reviewer_BGKL · 2025-10-31

**Soundness:** 3
**Presentation:** 2
**Contribution:** 2
**Rating:** 2
**Confidence:** 4

**Summary:**

This paper presents **RFMedSAM 2**, an automatic prompt refinement framework for **Segment Anything Model 2 (SAM 2)**, designed to enhance volumetric (3D and multi-slice) medical image segmentation performance. The method first explores the upper performance limit of SAM 2 through **custom fine-tuning adapters** and **ground-truth prompts**, achieving a Dice Similarity Coefficient (DSC) of **92.30%** on the BTCV dataset — a **12% improvement** over nnUNet. Then, an **Automatic Prompt Generation Module** is introduced, where a UNet autonomously predicts masks and bounding boxes as prompts for SAM 2. Within SAM 2, a **Dual-stage Refinement** mechanism iteratively improves the segmentation results.

**Strengths:**

1. The proposed automatic prompt generation and dual-stage refinement pipeline effectively reduce SAM’s dependence on manual prompting, moving toward fully automatic medical image segmentation.
2. The experimental validation is comprehensive, covering multimodal 3D medical datasets (CT/MRI), including BTCV and AMOS2022, and comparing against strong baselines such as nnUNet and MedSAM.
3. The method consistently surpasses current state-of-the-art models, achieving **+6.4% Dice improvement** on BTCV, demonstrating clear practical gains.

**Weaknesses:**

1. The core pipeline mainly relies on combining and fine-tuning existing models (SAM 2 and UNet) with limited theoretical novelty.
2. Details regarding the Bayesian search and dual-stage refinement — including parameter settings, loss functions, and training strategies — are insufficient, affecting reproducibility.
3. The combination of automatic prompting and dual-stage refinement in SAM 2 may introduce significant inference-time overhead, yet the paper does not report runtime or memory cost analysis.
4. The code is not open-sourced, which limits reproducibility and community verification.

**Questions:**

1. Can the UNet-generated prompts generalize across imaging modalities, and how does the system handle low-contrast or complex anatomical regions?
2. Please clarify whether the training and evaluation settings for nnUNet and MedSAM are strictly consistent to ensure fair comparison.

---

### Official Review · Reviewer_N4s2 · 2025-10-31

**Soundness:** 4
**Presentation:** 4
**Contribution:** 2
**Rating:** 2
**Confidence:** 4

**Summary:**

This paper extends the Segment Anything Model 2 (SAM 2) to the medical domain by introducing an automatic prompt generation and refinement framework. The authors first train a U-Net to generate initial masks and bounding boxes that serve as prompts for SAM 2. They further enhance SAM 2 with depth-wise convolutional adapters in the attention blocks, CNN-Adapters in the FPN module, and modifications to the frame selection and memory attention mechanisms. Evaluations on the BTCV and AMOS datasets demonstrate that the proposed method achieves leading performance, outperforming several strong automatic segmentation baselines as well as prompt-based SAM variants.

**Strengths:**

1. The paper has a clear motivation: it identifies SAM 2’s key limitations, including prompt dependency, binary mask outputs, and lack of semantic understanding, and systematically addresses them through architectural and training refinements.
2. The paper demonstrates solid engineering, with well-designed modules and thorough ablation studies that clearly support the claimed improvements.
3. The method achieves competitive performance, outperforming other SAM-based models on both BTCV and AMOS datasets when prompts are available.

**Weaknesses:**

1. All evaluations are conducted only on abdominal CT datasets, limiting the generalizability of the findings.
2. The overall framework heavily depends on the U-Net’s outputs for prompt generation, which makes it susceptible to error propagation—mistakes from the U-Net cannot be effectively corrected by the subsequent refinement stages.
3. Following 1 and 2, since the U-Net already performs better than many competing methods, it is unclear how much the proposed system contributes beyond the U-Net itself. Additional evaluations on more challenging datasets where the U-Net performs poorly would strengthen the claims.
4. The method’s performance is comparable to other automatic segmentation approaches despite its more complex architecture. A detailed analysis of computational cost and efficiency would help justify the added architectural complexity.

**Questions:**

1. Could the authors evaluate the method on more diverse datasets, particularly on harder modalities or anatomical regions where the U-Net baseline performs poorly, to better demonstrate generalization and robustness?
2. The novelty appears limited, as the improvements primarily stem from architectural and engineering refinements rather than fundamentally new algorithmic contributions. Could the authors clarify what conceptual advancements distinguish this work from prior SAM adaptation studies?
3. The paper is dense with implementation and ablation details, which may obscure the main contributions. The authors could consider simplifying or streamlining the presentation, or expanding the work into a journal version where the technical depth would be more appropriate.

---

### Official Review · Reviewer_aGLU · 2025-11-01

**Soundness:** 3
**Presentation:** 3
**Contribution:** 2
**Rating:** 4
**Confidence:** 4

**Summary:**

The paper proposes RFMedSAM 2, an extension of SAM 2 for medical image segmentation. The framework integrates an auxiliary UNet that automatically generates mask- and box-based prompts for SAM 2, enabling prompt-free segmentation. It further introduces depth-wise convolutional adapters and CNN-adapters to better capture spatial information while keeping the number of trainable parameters low. The model performs two refinement stages within SAM 2 to enhance accuracy. Experiments on AMOS22 and BTCV datasets show consistent gains over nnUNet and existing SAM-based methods.

**Strengths:**

- Provides a thorough empirical study of SAM 2’s behavior on medical images, including analysis of prompt dependency and temporal positioning.
- Well-designed engineering pipeline combining UNet and SAM 2, with clear modularity and reasonable computational cost.
- Extensive ablation studies support each design choice (adapters, frame selection, refinement, prompt generation).
- Achieves strong performance across two standard medical datasets, surpassing established baselines like nnUNet, SAMed, and SAM3D.
- Writing and figures are clear and comprehensive, making the method easy to follow and reproduce.

**Weaknesses:**

- Conceptual novelty is limited; the framework mainly adapts and combines existing components (UNet, SAM 2, adapters) without new algorithmic insight.
- The “automatic prompt refinement” is largely implemented via a separate UNet, which reduces the claimed autonomy of the system.
- Improvements over strong baselines (e.g., nnUNet) are modest on larger datasets and may not reflect a fundamental advance.
- Lack of theoretical or analytical understanding of why the multi-stage refinement contributes to performance.
- The evaluation uses only CT datasets; generalization to MRI or other modalities remains untested.

**Questions:**

- How sensitive is the system to errors from the UNet-generated prompts? Would the performance drop significantly if UNet predictions are noisy?
- How does the method perform in real-time or low-resource clinical settings, given SAM 2’s high memory requirements?

---

### Note · Authors · 2025-11-14

I have read and agree with the venue's withdrawal policy on behalf of myself and my co-authors.